# Parental effects alter the adaptive value of an adult behavioural trait

**Rebecca M Kilner[1]\*, Giuseppe Boncoraglio[1], Jonathan M Henshaw[2], Benjamin JM Jarrett[1], Ornela De Gasperin[1], Alfredo Attisano[1,3], Hanna Kokko[4]**

[1]Department of Zoology, University of Cambridge, Cambridge, United Kingdom; [2]Research School of Biology, Australian National University, Canberra, Australia; [3]Museum and Institute of Zoology, Polish Academy of Sciences, Warsaw, Poland; [4]Institute of Evolutionary Biology and Environmental Studies, University of Zürich, Zürich, Switzerland

**Abstract** The parents' phenotype, or the environment they create for their young, can have long-lasting effects on their offspring, with profound evolutionary consequences. Yet, virtually no work has considered how such parental effects might change the adaptive value of behavioural traits expressed by offspring upon reaching adulthood. To address this problem, we combined experiments on burying beetles (*Nicrophorus vespilloides*) with theoretical modelling and focussed on one adult behavioural trait in particular: the supply of parental care. We manipulated the early-life environment and measured the fitness payoffs associated with the supply of parental care when larvae reached maturity. We found that (1) adults that received low levels of care as larvae were less successful at raising larger broods and suffered greater mortality as a result: they were low-quality parents. Furthermore, (2) high-quality males that raised offspring with low-quality females subsequently suffered greater mortality than brothers of equivalent quality, which reared larvae with higher quality females. Our analyses identify three general ways in which parental effects can change the adaptive value of an adult behavioural trait: by influencing the associated fitness benefits and costs; by consequently changing the evolutionary outcome of social interactions; and by modifying the evolutionarily stable expression of behavioural traits that are themselves parental effects.

**\*For correspondence:**
rmk1002@cam.ac.uk

**Competing interests:** The authors declare that no competing interests exist.

## Introduction

Parental effects can play a major non-genetic role in determining an offspring's phenotype, either via the parents' phenotype or through the environment that parents create (*Badyaev and Uller, 2009*; *Wolf and Wade, 2009*). It is well-appreciated that parental effects can persist beyond the juvenile stage, profoundly influencing the offspring's subsequent adult phenotype, including key fitness-related traits such as lifespan (*Bateson et al., 2004*; *Nussey et al., 2007*) and fecundity (*Wilkin and Sheldon, 2009*; *Emlen et al., 2012*). Consequently, parental effects can change the nature and pace of ecological and evolutionary change, potentially allowing organisms to adapt quickly in a rapidly changing environment (e.g., *Räsänen and Kruuk, 2007*; *Badyaev and Uller, 2009*; *Pfennig and Martin, 2009*; *Duckworth et al., 2015*).

Here, we consider whether parental effects can also change the adaptive value of behavioural traits, something that has been rather overlooked in previous analyses. Yet, behavioural traits are among the first aspects of the phenotype to adapt to a changed environment (*West-Eberhard, 2003*; *Zuk et al., 2014*), and parental effects may play a key role in this process. We address this problem by focusing specifically on how conditions in early-life influence the fitness costs and benefits associated with the supply of parental care in adult life, using a combination of experiments and new theory.

**eLife digest** The burying beetle is an unusual insect in that both the father and the mother take care of their young larvae. They do this by providing food in the form of a small dead animal, such as a mouse, from which they diligently remove any fur or feathers, and by defending both the food and the larvae from rivals. These actions reduce the fitness of the parents, which can be estimated by measuring by how long they survive after caring for their brood. They also increase the health of the larvae, as measured by how large the larvae are when they move away from the carcass to pupate.

Kilner et al. wanted to know how the parenting received by larvae affects their behaviour when they grow up and have their own offspring. Larvae were given varying amounts of care, ranging from none at all to five days (which is the typical length of the larval stage for burying beetles). Larvae that received little or no care grew up to become low-quality parents, whereas those that received lots of care became high-quality parents. A low-quality parent is, by definition, a parent that becomes less fit as a result of rearing offspring; a high-quality parent providing the same amount of care would not suffer such a large reduction in its fitness.

Each of the female beetles from this first experiment was then mated with a high-quality male and together they took care of their offspring. Kilner et al. observed that the fathers lived longer when they were paired with high-quality mothers than they did when they were paired with lower quality mothers. This happened because the lower quality mothers effectively exploited the fathers, forcing them to do more of the parenting. Although the males gained by raising healthy larvae, they paid a price by dying at a younger age.

Results from these insect experiments are not directly linked to human behaviour, but they might tell us why animals of other species are generally so careful to choose a mate that matches them in quality. In this way, they can avoid being exploited when the pair work together to raise young. In future, Kilner et al. will investigate how beetles adjust their parenting effort in response to the effort put in by their partner: can they estimate parental quality directly, or do they simply observe how much care the other partner is providing?

By choosing to focus on parental care, we are able to analyze three different ways in which parental effects might hypothetically influence the adaptive value of any behavioural trait.

First, by focusing on uniparental care, we can investigate how parental effects change the relative fitness payoffs associated with a behaviour performed in adulthood. We assess these fitness payoffs experimentally by measuring the fitness gained from supplying care (the benefit of care) as well as the fitness simultaneously lost (the cost of care, e.g., *Grafen, 1984*).

The second approach takes advantage of the fact that parental care is often supplied by both parents, working as a team. Parents commonly respond to the effort put in by their partner (e.g., *Johnstone and Hinde, 2006*), which means that the parental behaviour exhibited by one parent can potentially change the fitness of its partner by inducing a change in parental effort (e.g., *Houston and Davies, 1985*; *Houston et al., 2005*). This allows us to analyze how parental effects experienced in early life might influence the outcome of a social interaction in adulthood.

For a social behaviour, such as biparental care, the relative magnitude of the fitness costs and benefits of the interaction with another individual are key to understanding whether the outcome is cooperative or exploitative (e.g., *West et al., 2006*). If the fitness of both social partners is enhanced by the interaction then the outcome is cooperative, but if one party gains fitness at the other's expense then it is selfishly exploiting its social partner. For biparental care, cooperation and selfish exploitation are each possible elements of parents working together to raise young. A cooperative element is possible when the two parents have an equal genetic stake in their shared young and so derive equal fitness benefits from the supply of care. But, a selfish exploitative element is also possible due to conflict between the parents over how to divide the fitness costs associated with the supply of care (*Arnqvist and Rowe, 2005*; *Houston et al., 2005*; *Lessells, 2006*). Selection favours parents that force their partner to sustain a greater fitness cost which, in accordance with the definition of sexual conflict (*Kokko and Jennions, 2014*), means that the parent bearing the increased cost of care is selfishly exploited by its partner.

Parental effects, experienced in early life, can potentially change the outcome of interactions during the supply of biparental care in adulthood by changing the costs and benefits associated with

the supply of care (*Barta et al., 2002*; *Harrison et al., 2009*; *Lessells and McNamara, 2012*). Theoretical analyses suggest that offspring that develop in a well-resourced environment may mature into high-quality parents. In this context, the term 'high-quality' is precisely defined: for the same provision of care, a 'high-quality' parent sustains a lower fitness cost than a 'low-quality' parent (*Lessells and McNamara, 2012*). In theory, a high-quality parent is vulnerable to selfish exploitation by a low-quality partner who is incentivized to reduce its greater costs of care by offloading them onto the superior quality parent (*Lessells and McNamara, 2012*). Whether this ever happens unknown (but see *Monaghan et al., 2012*).

A third feature of parental care is that it directly influences features of the early-life environment in which the next generation develops and thus directly induces corresponding change in the offspring's phenotype: in other words, and in common with several other behavioural traits, parental care is itself a type of parental effect (*Badyaev and Uller, 2009*; *Wolf and Wade, 2009*). Thus, early-life conditions potentially change the costs and benefits of parental care in adulthood, which in turn influence the early-life conditions experienced by the next generation. The influence of these transgenerational effects on the evolution of parental care strategies is, however, unknown. We develop new theoretical models, informed by our experiments, to analyze how these long-term fitness consequences of parental effects might feed back to change optimal levels of parental care.

Our experiments use the burying beetle, *Nicrophorus vespilloides,* as a model organism. Like other *Nicrophorus* beetles, this species uses the carcass of a small vertebrate for reproduction. It exhibits unusually elaborate biparental care, though one parent (of either sex) can raise offspring singlehandedly (e.g., *Ward et al., 2009*). A major advantage of using the burying beetle is that we can measure correlates of fitness that are associated with receiving and supplying parental care. Indeed, our study differs from most previous empirical work on parental care by measuring fitness, rather than by quantifying behaviour itself. The rationale underpinning our experimental approach is this: behavioural traits associated with care cannot straightforwardly be mapped onto fitness (*Sheldon, 2002*; *Harrison et al., 2009*)—yet, measures of fitness are key to understanding evolution of parental care (*Sheldon, 2002*; *Harrison et al., 2009*). By measuring changes in fitness associated with the supply and receipt of parental care, rather than quantifying behavioural traits, we can therefore more easily use our experiments to draw evolutionary conclusions.

## Materials and methods

### Study species: the burying beetle *N. vespilloides*
#### Parental care
Reproduction in the burying beetle centres upon the body of a small dead vertebrate (*Scott, 1998*). Having located a carcass, males and females each contribute extensively to parental care, and each can carry out all the diverse tasks associated with caring for offspring if widowed (e.g., *Smiseth et al., 2005*; *Smiseth et al. 2006*; *Suzuki and Nagano, 2009*), or in the case of females, if they alone locate a carcass (*Müller et al., 2007*). Importantly, when both parents are present, path analyses of observational data show that contributions to care by one parent are negatively related to the effort of the partner (e.g., *Jenkins et al., 2000*; *Walling et al., 2008*), which is consistent with some sort of compensatory response to the care provided by the partner.

Together, the two parents remove any fur or feathers from the carcass, roll the flesh into a ball, smear it with antimicrobials, and bury it in a shallow grave. The eggs hatch in the soil nearby, and the larvae crawl to the carcass. There, they solicit food from their attendant parents and consume the flesh themselves (*Smiseth et al., 2003*). Parents also defend the offspring and carcass from potential rivals (*Scott, 1998*). Roughly, 5 days after hatching, larvae disperse from the carcass to pupate in the soil. At this point, or in the days beforehand, parents fly off in search of fresh carrion. Larvae are capable of survival without any post-hatching parental care at all (*Eggert et al., 1998*; *Smiseth et al., 2003*; *Schrader et al., 2015*), which means the quality of the early-life environment can be manipulated simply by removing parents at different intervals after hatching (e.g., *Boncoraglio and Kilner, 2012*).

### Quantifying fitness costs and benefits associated with care
In the burying beetle, all forms of care incur measurable fitness costs and benefits (e.g., *Ward et al., 2009*; *Cotter et al., 2010*). We quantified the fitness benefits of care by measuring larval mass at

dispersal because this is strongly correlated with survival (e.g., *Eggert et al., 1998*; *Lock et al., 2004*). We used lifespan to measure the fitness costs associated with parental care, because several aspects of the burying beetle's reproductive biology mean that a longer lifespan contributes directly to greater fecundity in each sex and is therefore a good proxy for fitness. For example, in nature, *N. vespilloides* is an opportunistic breeder because it is reliant on locating small carrion to produce offspring (*Schwarz and Müller, 1992*; *Scott, 1998*). The longer it lives, the more likely it is to find this key resource. Males that are unsuccessful at locating carrion may attempt to attract females for mating by secreting pheromones (*Eggert and Müller, 1989a*, *1989b*; *Eggert, 1992*). Females can store sperm from these matings for use in future reproduction (*Müller and Eggert, 1989*) and males commonly gain paternity without attending the carcass upon which larvae are raised (*Müller et al., 2007*). Therefore, the longer a male lives, the more likely he is to be successful in acquiring matings and increasing his reproductive success. In our previous laboratory experiments, we have shown that a longer lifespan enhances lifetime reproductive success in both sexes by affording a greater number of opportunities for reproduction (*Ward et al., 2009*; *Cotter et al., 2010*, *2011*). Note that there is no difference between the sexes in the lifespan of virgins (*Figure 1—figure supplement 1*).

## *N. vespilloides* colony and housing conditions

All the individuals used in this experiment belonged to a captive colony (kept at a constant temperature: 21°C, with a 16-hr:8-hr light:dark cycle) established at Cambridge University in 2005 and supplemented every summer thereafter with wild caught adults collected under licence from local field sites at Byron's Pool and Wicken Fen in Cambridgeshire. Adults were housed individually in plastic boxes (12 × 8 × 2 cm) filled with moist soil and fed twice a week with ca. 1 g minced beef. For breeding, pairs of unrelated individuals were placed into larger plastic boxes (17 × 12 × 6 cm) half filled with moist soil, provided with a 15–35 g freshly thawed mouse carcass and kept in the dark to simulate natural underground conditions. The larvae disperse from the carcass to pupate roughly eight days after pairing. Pupation takes approximately three weeks and sexual maturity is reached approximately two weeks after eclosion.

## Experiment 1: the influence of parental effects on the costs and benefits of parental care provided in adult life

This experiment spanned two generations. In Generation 1, we manipulated larval developmental environment via a parental effect. We then kept these larvae to breed as adults in Generation 2, when we measured the fitness costs and benefits associated with providing care.

### Generation 1—manipulation of larval developmental environment via a parental effect

We paired virgins when they were 2–3 weeks old and allowed them to breed under standard conditions on a dead mouse. To minimize confounding effects in our manipulations, we subsequently removed all males from the breeding boxes ca. 53 hr after pairing. By this point, egg laying and carcass preparation were complete, but the larvae had yet to hatch (in our colony, this starts 71.28 hr $\pm$ 1.47 SE after pair formation; $n = 47$ pairs checked every ca. five hours 55–96 hr after pairing). We then randomly assigned each brood to one of two maternal care treatments: either widowed females were removed from their breeding box around hatching (roughly 71 hr after pairing) or removal occurred 24 hr after hatching (roughly 95 hr after pairing). In previous work, we found that larvae that receive maternal care for the first 24 hr after hatching have a similar mass and subsequent lifespan as larvae that receive full-time care by both parents (*Boncoraglio and Kilner, 2012*). Carcass mass did not differ between care duration treatments ($F_{1, 58} = 0.02$, p = 0.960). In both treatments, we counted the number of dispersing larvae in each family, and weighed the brood at this point, before transferring larvae to new boxes for pupation and eclosion. This protocol was repeated over four weekly batches. Consistent with previous studies (*Eggert et al., 1998*; *Boncoraglio and Kilner, 2012*), the duration of maternal care positively affected final brood mass and size as well as individual offspring mass, both as a larva (calculated as average body mass at dispersal: total brood mass divided by number of dispersing larvae) and as an adult (pronotum width; for each measure of body size, one-way ANOVA (Analysis of Variance), p < 0.045, $n = 59$ broods). Nevertheless, the range in body size we generated experimentally did not differ significantly from the range in size in field-caught beetles (see *Figure 1—figure supplement 2*).

## Generation 2—measuring the costs and benefits of parental care

Two weeks after eclosion of the larvae from Generation 1, we randomly selected two females and two males per family from each treatment, until we obtained 40 experimental individuals altogether (20 of each sex). Each was then paired with a young unrelated virgin from the laboratory stock population and given a 15–25 g mouse carcass to start breeding. Whenever possible, tetrads of siblings from the same family in Generation 1 were matched with tetrads of siblings from a stock family, yielding four pairs in total per tetrad. Stock individuals were removed 53 hr after pairing so we could measure the fitness costs and benefits associated with the provision of post-hatching care by single parents, in the absence of any potential confounding partner effects.

To eliminate any potential confounding pre-hatching effects on larval quality, we cross-fostered the larvae to be raised by our experimental subjects, a manipulation facilitated by the inability of parents to recognize their own offspring directly (*Scott, 1998*). At hatching, we moved the experimental individuals and their carcass to a new breeding box. Hatchlings and unhatched eggs remained in the box and were fostered out to other experimental subjects for post-hatching care. Foster larvae were drawn at random from individuals breeding in the same batch and were formed into broods of either 5 larvae or 20 larvae. These brood sizes were chosen because they were representative of the range in brood size we had recorded in our colony in previous generations. In this generation of our stock population, for example, brood size at dispersal in stock broods ranged from 3 to 24. Within each brood size treatment, foster broods did not differ in mass at hatching (always p > 0.605). At dispersal, we measured total brood size and mass in each brood. At this point, adults were collected and housed alone under standard conditions until their time of death, and their lifespan was recorded. Overall, our experiment used 160 experimental individuals (i.e., 20 individuals per maternal care duration treatment per brood size treatment; 80 in total per sex) drawn from 40 different families. Carcass mass at breeding did not differ among the four treatments (one-way ANOVA, always p > 0.240, *n* = 160 breeding bouts). One individual (a 0 hr care female, given 20 larvae in adulthood) rejected the experimental brood and was excluded from subsequent analyses.

## Statistical methods

The data we collected and analyzed are archived in Dryad (*Kilner et al., 2015*). We used linear mixed models (LMMs) except for the analyses of brood size at dispersal, where generalized LMMs (GLMMs) with a Poisson error distribution were employed because the data were not normally distributed. The R package lme4 was used (*Bates et al., 2014*). The variances in parenting performance in small and large broods differed significantly (Levene test, $F_{1,\ 77}$ = 14.41, p < 0.001), so we square-root transformed the data prior to analysis. The experimental data for males and females were analyzed separately, but using the same approach. All models included the family of origin of the experimental individuals and the family of the donor larvae nested within batch as random intercept effects. Parental care duration (0 hr, 24 hr) and brood size (small = 5 larvae, large = 20 larvae) treatment effects were tested by two-level fixed factors. When present, carcass mass, brood mass at dispersal, and adult size were entered as covariates. Longevity data were analyzed using a Cox's proportional hazard model with the parent's family nested within batch using the R package coxme (*Therneau, 2012*). All analyses were run with R 3.1.2 (*R Development Core Team, 2013*).

## Experiment 2—the influence of parental effects on the outcome of a social interaction

This experiment also spanned two generations. In Generation 1, we manipulated larval developmental environment via a parental effect. We then kept only female larvae to breed as adults in Generation 2. Females were paired with males that had received full-time care as larvae and the pair was allowed to raise offspring together. After reproduction, we measured the fitness costs and benefits associated with providing care for both sexes.

### Generation 1—manipulation of larval developmental environment via a parental effect

The parents of experimental females were paired as 2- to 3-week-old virgins and given a mouse carcass for breeding. As before, all males were removed from the breeding boxes ca. 53 hr later, just prior to larval hatching. Broods were then randomly assigned one of four care duration treatments: 0 hr care, with widowed females being removed from their breeding box around hatching (71 hr after

pairing); 8 hr care (widowed female removed 79 hr after pairing); 24 hr care (widowed female removed 95 hr after pairing); or full care/192-hr care (widowed female allowed to rear the brood alone until dispersal of the offspring from the carcass). Stock pairs, where both parents were allowed to rear their brood until larval dispersal, were also maintained alongside these treatments. At dispersal, we measured total brood mass and size and set up the larvae individually for pupation and eclosion. Carcass mass did not differ among care duration treatments ($F_{3, 117} = 0.388$, p = 0.762). This protocol was repeated over four batches, initiated weekly.

As before, we found that the duration of care positively affected final brood mass and size as well offspring mass, both as a larva (again measured as average larval mass at dispersal) and as an adult (pronotum width; for each measure of body size, one-way ANOVA, p < 0.003, n = 121 broods). These findings are consistent with previous studies (*Eggert et al., 1998*; *Boncoraglio and Kilner, 2012*).

### Generation 2—measuring the outcome of a social interaction

Two weeks after eclosion of the experimental females, we randomly selected two sisters per brood per care duration treatment. Each batch yielded five pairs of sisters per treatment giving us 40 experimental females in total. With 4 batches, we had 160 experimental females altogether, manipulated along a gradient in their capacity to provide care.

These experimental females were each paired with a young unrelated virgin male from the stock population and given a 15–25 g mouse carcass to start breeding. Males were drawn from tetrads of brothers from the same stock family, with each member of the tetrad distributed across the four quality treatments experienced by the females. The 160 males in this experiment were thus derived from 40 different families. Pairs were allowed to rear their brood together until larval dispersal. We measured average larval mass at dispersal and removed the adults at this point, temporarily housing females and males individually under standard conditions. Then, 14 days after the start of their first breeding bout, we performed a second breeding bout using the same protocol as described above, and using the same pairs as for the first breeding bout (i.e., same partners in both rounds). After dispersal of the second brood, experimental females and their partners were kept separately under standard conditions until their time of death. We recorded their lifespan. We allowed pairs to reproduce twice in this experiment, because burying beetles are bivoltine in nature (*Schwarz and Müller, 1992*) and breeding pairs twice increased the likelihood of detecting an effect of female quality on male fitness. Four stock males and one experimental female died before starting their second breeding bout. Males were replaced with other stock partners, which were not included in the data set, while the female was not replaced. Carcass mass at first or second breeding did not differ among experimental subgroups (one-way ANOVA, p > 0.440, n = 319 breeding bouts).

## Statistical methods

The data we collected and analyzed are archived in Dryad (*Kilner et al., 2015*). For data that were not normally distributed, we used GLMMs assuming a Poisson error distribution, and using the R package lme4 (*Bates et al., 2014*). A LMM was used for analyzing average offspring mass, because these data were normally distributed. All models included batch number as a random factor and both the female's and male's family of origin nested within batch as random intercept effects. Maternal care duration (0 hr, 8 hr, 24 hr, 192 hr/full care) was entered as a four-level fixed factor. Where relevant, carcass mass, female and male adult size were entered as covariates. In the models of reproductive output, we analyzed data collected over the two experimental breeding bouts. Analyses of brood size used the total number of offspring produced over both breeding attempts. For average larval mass, we calculated this value separately for each breeding attempt and then derived the mean. (The results for each measure of offspring performance are qualitatively similar when data from the first breeding bout alone are analyzed.) Longevity data were analyzed using a Cox's proportional hazards model with the family of the experimental individual nested within batch using the R package coxme (*Therneau, 2012*). Post hoc analyses were performed adopting the Bonferroni correction for multiple testing. All analyses were performed with R 3.1.2 (*R Development Core Team, 2013*).

## Model: sexual conflict and the transgenerational impact of parental effects

To provide further insight into our empirical results, we constructed a mathematical model of the evolution of parental care in an idealized population of burying beetles (*Figure 1*). Our model

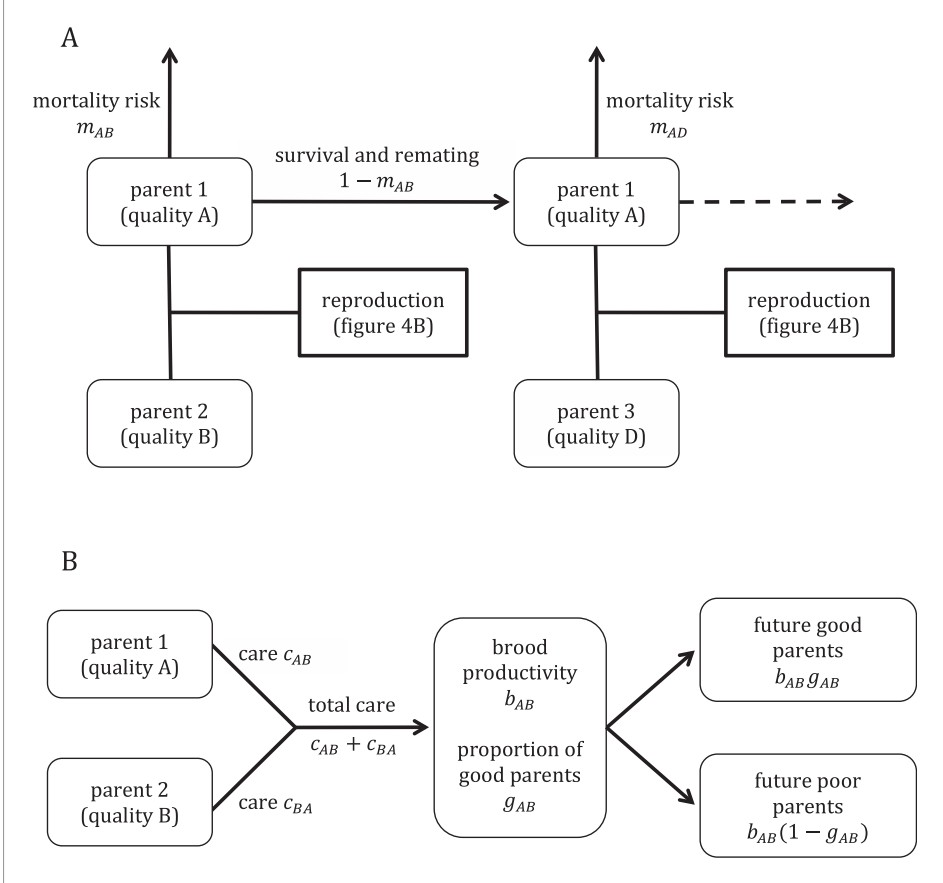

**Figure 1**. Overview of the model structure, including mortality and survival (**A**) and reproduction (**B**).
The following figure supplements are available for figure 1:

**Figure supplement 1**. Longevity of adult virgin males ($n = 43$) and females ($n = 50$) (**A**) shown as a cumulative survival plot and (**B**) comparing mean ± S.E.M lifespan for each sex.
**Figure supplement 2**. Frequency distribution of the size of field-caught *N. vespilloides* (shown in blue) and the experimental *N. vespilloides* described in this study (shown in red).

explores how individual variation in parental quality affects the amount of care that parents evolve to provide. We assume that the care offspring receive during development affects their own future quality as parents, a type of parental effect. To explore the impact of sexual conflict on care decisions, we compare predicted levels of care when coparents are unrelated to each other and when they are genetically identical (i.e., $r = 1$ between coparents). The latter scenario entirely removes sexual conflict over care decisions. We first provide a general outline of the model before considering each piece in more detail.

## Outline of model

We assume a large, well-mixed population that breeds continuously. Adult beetles alternate their time between searching for suitable corpses, waiting at corpses for a mate to arrive, and breeding. When a searching individual locates an unoccupied corpse, or one that is occupied only by an opposite-sex adult, it remains there to breed. Corpses that are occupied by a same-sex competitor are avoided. This means that the first female and the first male to encounter a corpse form a monogamous breeding pair. They care jointly for their offspring, and once their brood reaches maturity, the pair

separates and each beetle resumes its search for corpses. Although breeding is continuous in our model, we do not specify how much time beetles spend engaged in each activity. Rather, we model trade-offs between care provision, fecundity, and mortality that do not depend explicitly on time.

Adult beetles vary in their quality, with higher quality individuals suffering lower mortality costs after providing any given level of parental care. For simplicity, we divide the adult population into 'good' and 'poor' parents. Parental quality is determined during development and persists throughout an adult's life. We assume that opportunities to develop into a good parent are limited at the population level by competition for resources. The overall proportion of larvae that develop into good parents is consequently fixed, regardless of the average level of care in the population. This is analogous to density-dependent effects on survival.

Care has two effects on offspring. First, the total number of larvae surviving from a brood ('brood productivity') increases with the sum of care provided by the two parents. Second, larvae from broods receiving high levels of care are more likely to develop into good parents than those receiving low levels of care.

The level of care an individual provides can vary according to both its own parental quality (good or poor) and that of its mate. However, individuals do not respond plastically to the immediate level of care that their mate provides. This is equivalent to a 'sealed bid' model of parental care (*Houston and Davies, 1985*), though in our case, the bid is multidimensional (quality-dependent). We also assume that there are no sex differences in the fitness costs and benefits of caring, so that males and females provide the same levels of care.

We write $c_{AB}$ for the amount of care provided by a parent of type $A$ that is paired with a parent of type $B$. Each of the subscripts can equal either $G$ (for good parents) or $P$ (for poor parents) to give us a total of four possible care levels: $c_{GG}$, $c_{GP}$, $c_{PG}$, and $c_{PP}$. For simplicity, we assume a haploid genetic system, where each care level is coded for by a single allele. We allow each care level to evolve independently of the others.

We begin by calculating the reproductive value of good and poor parents in a population where every individual behaves in the same way. We then calculate reproductive values of mutants whose care behaviour differs from the population norm. This allows us to define selection gradients on the amount of care provided in different circumstances, which we then use to find evolutionarily stable levels of care. We repeat this analysis for populations in which coparents are genetically identical in order to explore the effects of sexual conflict over parental care.

## Mortality

For simplicity, we assume that mortality of adult beetles occurs during the search phase between bouts of reproduction. An individual's probability of mortality increases with the level of care it provided in the previous bout. Mortality also depends on an individual's quality as a parent, with higher quality individuals suffering from lower mortality after providing any given level of care.

We write $m_G(c)$ and $m_P(c)$ for the mortality risk of a good parent and a poor parent, respectively, after providing care of $c$. We assume that mortality risk increases as a convex function of the amount of care a parent provides (see Appendix 1). By definition, good parents suffer lower mortality than poor parents after providing the same level of care, so we assume $m_G(c) < m_P(c)$ for all $c$. The probability that a parent of type $A$ dies between reproductive bouts after raising offspring with a parent of type $B$ can be written as

$$m_{AB} = m_A(c_{AB}). \qquad (1)$$

Now, let $g_1$ be the (assumed fixed) proportion of newly mature adults that are good parents and let $g_2$ be the proportion of all mating adults that are good parents. These two quantities might not be equal due to differences in mortality between good and poor parents. Since we assume continuous breeding, $g_2$ is constant with time.

When a good parent reproduces, the probability that its mate is also a good parent is $g_2$, in which case, the focal parent faces a subsequent mortality risk of $m_{GG}$. Similarly, the probability that the focal parent's mate is a poor parent is $1 - g_2$, in which case, the mortality risk is $m_{GP}$. The overall probability that a good parent dies between reproductive bouts is thus

$$m_G = g_2 m_{GG} + (1 - g_2) m_{GP}. \qquad (2)$$

For poor parents, the average mortality risk is similarly

$$m_P = g_2 m_{PG} (1 - g_2) m_{PP}. \tag{3}$$

Good parents make up $g_1$ of newly mature adults and mate on average $1/m_G$ times in their lives, while poor parents make up $1 - g_1$ of new adults and mate $1/m_P$ times. The proportion of mating adults that are good parents is therefore

$$g_2 = \frac{g_1/m_G}{g_1/m_G + (1 - g_1)/m_P}. \tag{4}$$

*Equation 2* through *Equation 4* can be solved simultaneously to yield values for $g_2$, $m_G$, and $m_P$ in terms of $g_1$.

## Effect of care on brood productivity

The average number of offspring surviving from a brood depends on the total care provided by the two parents. For a mating between parents of types A and B, we write the expected brood productivity as

$$b_{AB} = b(c_{AB} + c_{BA}). \tag{5}$$

We assume that brood productivity is a concave function of the amount of care received (see Appendix 1).

In order to simplify calculations, we normalize brood productivity so that an average adult has lifetime reproductive success of one. This normalization is for mathematical convenience and does not affect the generality of our results. We calculate pre-normalized average lifetime reproductive success as follows. First, each good parent mates on average $1/m_G$ times during its life. A proportion $g_2$ of these matings are with good parents and result in brood productivity of $b_{GG}$. The remaining proportion $1 - g_2$ are with poor parents and result in brood productivity of $b_{GP}$. The average lifetime reproductive success of a good parent is thus $(1/m_G)[g_2 b_{GG} + (1 - g_2)b_{GP}]$. Similarly, for poor parents, the average lifetime reproductive success is given by $(1/m_P)[g_2 b_{GP} + (1 - g_2)b_{PP}]$. Since good parents make up a proportion $g_1$ of newly mature adults, overall average lifetime reproductive success in the population is

$$\overline{b} = \left(\frac{g_1}{m_G}\right)[g_2 b_{GG} + (1 - g_2)b_{GP}] + \left(\frac{1 - g_1}{m_P}\right)[g_2 b_{GP} + (1 - g_2)b_{PP}]. \tag{6}$$

Normalized brood productivity can then be calculated as

$$b'_{AB} = \frac{b_{AB}}{\overline{b}}. \tag{7}$$

These calculations also lead to simple expressions for the proportions of newly mature offspring that were raised by parents of each possible combination of parental quality (i.e., both good, one good and one poor, or both poor). These are

$$q_{GG} = \left(\frac{g_1}{m_G}\right) g_2 b'_{GG}, \tag{8}$$

$$q_{GP} = \left[\left(\frac{g_1}{m_G}\right)(1 - g_2) + \left(\frac{1 - g_1}{m_P}\right)g_2\right] b'_{GP}, \tag{9}$$

$$q_{PP} = \left(\frac{1 - g_1}{m_P}\right)(1 - g_2) b'_{PP}. \tag{10}$$

Note that $q_{GG} + q_{GP} + q_{PP} = 1$.

## Transgenerational effects of care on future parental quality

We now determine the proportion of larvae raised by each type of pair that develops into good parents. We assume that opportunities to develop into a good parent are limited at the population level by access to resources and that surviving larvae compete for a fixed number of such opportunities. An individual's relative competitiveness is an increasing function of the total amount of care it received during development. We write the relative competitiveness of individuals raised by parents of types $A$ and $B$ as

$$f_{AB} = f(c_{AB} + c_{BA}). \tag{11}$$

Like brood productivity, we take competitiveness to be a concave function of the total care a brood receives (see Appendix 1).

Suppose that opportunities to develop into a good parent arise continuously at a constant rate. We assume that each newly mature adult competes for such opportunities for a fixed time $T$, during which its probability of gaining any particular opportunity is proportional to its competitiveness with some constant of proportionality $k > 0$. This means that for an individual with parents of types $A$ and $B$, the probability of success can be written as

$$g_{AB} = 1 - e^{-kf_{AB}T}. \tag{12}$$

By manipulating the above equation for $g_{GG}$, $g_{GP}$, and $g_{PP}$, we obtain the relationships

$$(1 - g_{GG})^{1/f_{GG}} = (1 - g_{GP})^{1/f_{GP}} = (1 - g_{PP})^{1/f_{PP}}. \tag{13}$$

Further, since the final overall proportion of good parents is fixed at $g_1$, we must also have

$$g_{GG}q_{GG} + g_{GP}q_{GP} + g_{PP}q_{PP} = g_1. \tag{14}$$

We can solve *Equation 13* and *Equation 14* simultaneously using numerical methods to yield values for $g_{GG}$, $g_{GP}$, and $g_{PP}$. Note that the solution does not depend on the choice of $k$ or $T$.

## Reproductive value of typical good and poor parents

We now have all we need to calculate the reproductive value of good and poor parents in a population where care behaviour is uniform. We write $v_G$ for the reproductive value of a good parent and $v_P$ for a poor parent. When a good parent mates with another good parent, their normalized brood productivity is $b'_{GG}$ Of this brood, a proportion $g_{GG}$ of offspring will develop into good parents themselves, while $1 - g_{GG}$ will develop into poor parents. The reproductive value that a good parent obtains from mating with another good parent is thus $\frac{1}{2}b'_{GG}[g_{GG}v_G + (1 - g_{GG})v_P]$. The factor of one half here accounts for the relatedness between an individual and its offspring. Similarly, good parents obtain reproductive value of $\frac{1}{2}b'_{GP}[g_{GP}v_G + (1 - g_{GP})v_P]$ from mating with a poor parent.

Since good parents mate on average $1/m_G$ times, and a proportion $g_2$ of these matings are with other good parents, the total reproductive value of a good parent is given by

$$v_G = \frac{1}{2m_G}\left\{g_2 b'_{GG}[g_{GG}v_G + (1 - g_{GG})v_P] + (1 - g_2)b'_{GP}[g_{GP}v_G + (1 - g_{GP})v_P]\right\}. \tag{15}$$

Similarly, the reproductive value of a poor parent is

$$v_P = \frac{1}{2m_P}\left\{g_2 b'_{GP}[g_{GP}v_G + (1 - g_{GP})v_P] + (1 - g_2)b'_{PP}[g_{PP}v_G + (1 - g_{PP})v_P]\right\}. \tag{16}$$

*Equation 15* and *Equation 16* can be rewritten as a matrix equation

$$\begin{bmatrix} v_G \\ v_P \end{bmatrix} = M \begin{bmatrix} v_G \\ v_P \end{bmatrix}, \tag{17}$$

where the entries of $M$ are given by

$$M_{11} = \frac{1}{2m_G} \left[ g_2 b'_{GG} g_{GG} + (1-g_2) b'_{GP} g_{GP} \right]$$

$$M_{12} = \frac{1}{2m_G} \left[ g_2 b'_{GG} (1-g_{GG}) + (1-g_2) b'_{GP} (1-g_{GP}) \right]$$

$$M_{21} = \frac{1}{2m_P} \left[ g_2 b'_{GP} g_{GP} + (1-g_2) b'_{PP} g_{PP} \right]$$

$$M_{22} = \frac{1}{2m_P} \left[ g_2 b'_{GP} (1-g_{GP}) + (1-g_2) b'_{PP} (1-g_{PP}) \right].$$

(18)

The reproductive value of good and poor parents can then be found by taking the right eigenvector of $M$ that corresponds to the eigenvalue 1.

### Evolutionarily stable levels of biparental care

We now have general expressions for the reproductive value of typical individuals in the population. We can similarly derive the reproductive values of mutants that differ in their care behaviour from population norms (see Appendix 1). We can use these to calculate selection gradients on the level of care provided in each of the four possible pairings of parental quality.

We write $c = (c_{GG}, c_{GP}, c_{PG}, c_{PP})$ for the vector of population values for each care level and we write $\widehat{c}$ for the corresponding vector for mutants. The selection gradient on $c$ is then given by

$$S(c) = \left( \frac{\partial \widehat{v}_G}{\partial \widehat{c}_{GG}}, \frac{\partial \widehat{v}_G}{\partial \widehat{c}_{GP}}, \frac{\partial \widehat{v}_P}{\partial \widehat{c}_{PG}}, \frac{\partial \widehat{v}_P}{\partial \widehat{c}_{PP}} \right) \bigg|_{\widehat{c}=c},$$

(19)

where $\widehat{v}_G$ and $\widehat{v}_P$ are the reproductive values of good and poor quality mutants, respectively.

We located evolutionarily stable levels of care by starting with arbitrary levels of care $c = c_0$ and then following the selection trajectories defined by $S$ until care levels converged to an equilibrium. This was done by iterating the equation

$$c_{t+1} = c_t + \Delta S(c_t),$$

(20)

with $\Delta$ a small positive constant (we found $\Delta = 0.01$ suitable). We checked informally for multiple equilibria by running the model with many widely spaced starting vectors $c_0$ to confirm that these starting vectors converged to the same equilibrium. Multiple equilibria were not found for any the parameter values we considered.

To assess the effect of sexual conflict on parental care, we compared equilibrium care levels under the above model to a hypothetical scenario that removes all sexual conflict by assuming that coparents are genetically identical (see Appendix 1).

## Results

### Experiment 1: the influence of parental effects on the costs and benefits of parental care

We found that parents that received no post-hatching care as larvae matured into low-quality mothers and fathers (using *Lessells and McNamara's (2012)* definition of parental quality).

#### Mothers

We found that females that received no parental care after hatching were less effective at providing care for broods of 20 larvae, which attained a lower mass by the time of dispersal from the carcass, when compared with broods raised by females that had received 24-hr care as larvae (*Table 1a*). These females also paid a greater fitness cost for providing lower quality care, suffering a shortened lifespan as a result (*Table 1b*; *Figure 2*). No equivalent differences in maternal quality were apparent when females were given only 5 larvae to raise ($X^2 = 0.65$, d.f. = 1, p = 0.42; *Figure 2*).

**Table 1**. Results from Experiment 1: the influence of parental effects on the costs and benefits of parental care provided in adult life

**Female-only care**

|  | Estimate | Standard error | t value | p value |
|---|---|---|---|---|
| a. Effect on brood mass (benefit of care) | | | | |
| Intercept | 0.07294 | 0.70654 | 0.103 | – |
| Duration of care as larva | −0.01430 | 0.09597 | −0.149 | 0.010 |
| Brood size raised as adult | 2.41542 | 0.08689 | 27.797 | <0.0001 |
| Carcass mass | 0.01451 | 0.01609 | 0.902 | 0.345 |
| Female pronotum | 0.12663 | 0.12270 | 1.032 | 0.282 |
| Duration of care as larva x brood size raised as adult | 0.44623 | 0.12150 | 3.673 | 0.0004 |

|  | Coefficient | Standard error | z value | p value |
|---|---|---|---|---|
| b. Effect on maternal survival (cost of care) | | | | |
| Duration of care as larva | 0.359 | 0.342 | 1.05 | 0.290 |
| Brood size raised as adult | 0.853 | 0.354 | 2.41 | 0.016 |
| Carcass mass | 0.059 | 0.059 | 1.00 | 0.320 |
| Female pronotum | 0.135 | 0.453 | 0.30 | 0.770 |
| Duration of care as larva x brood size raised as adult | −1.286 | 0.507 | −2.54 | 0.011 |

**Male-only care**

|  | Estimate | Standard error | t value | p value |
|---|---|---|---|---|
| c. Effect on brood mass (benefit of care) | | | | |
| Intercept | 0.60553 | 0.25415 | 2.383 | – |
| Duration of care as larva | 0.04445 | 0.04307 | 1.032 | 0.0002 |
| Brood size raised as adult | 0.75672 | 0.03818 | 19.820 | <0.0001 |
| Carcass mass | 0.01880 | 0.00572 | 3.287 | 0.001 |
| Male pronotum | −0.01450 | 0.04675 | −0.310 | 0.785 |
| Duration of care as larva x brood size raised as adult | 0.20190 | 0.05406 | 3.735 | 0.0002 |

|  | Coefficient | Standard error | z value | p value |
|---|---|---|---|---|
| d. Effect on paternal survival (cost of care) | | | | |
| Duration of care as larva | 0.183 | 0.352 | 0.52 | 0.600 |
| Brood size raised as adult | 0.686 | 0.334 | 2.05 | 0.040 |
| Carcass mass | 0.076 | 0.053 | 1.42 | 0.160 |
| Male pronotum | −0.448 | 0.385 | −0.12 | 0.910 |
| Duration of care as larva x brood size raised as adult | −0.856 | 0.474 | −1.81 | 0.071 |

Parental effects were created experimentally by exposing experimental subjects to either 0 hr or 24 hr of post-hatching care as larvae. They were then kept until adulthood and given broods of either 5 or 20 cross-fostered larvae to raise as a single parent. Their lifespan thereafter was recorded. Further details are given in the 'Materials and methods'.

## Fathers

Fathers that had experienced no post-hatching care also produced a lighter brood when given 20 larvae to rear (*Table 1c*, *Figure 3A*) compared with those that were provisioned after hatching. Again, these differences were not evident when males were given a brood of five to raise ($X^2 = 0.67$, d.f. = 1,

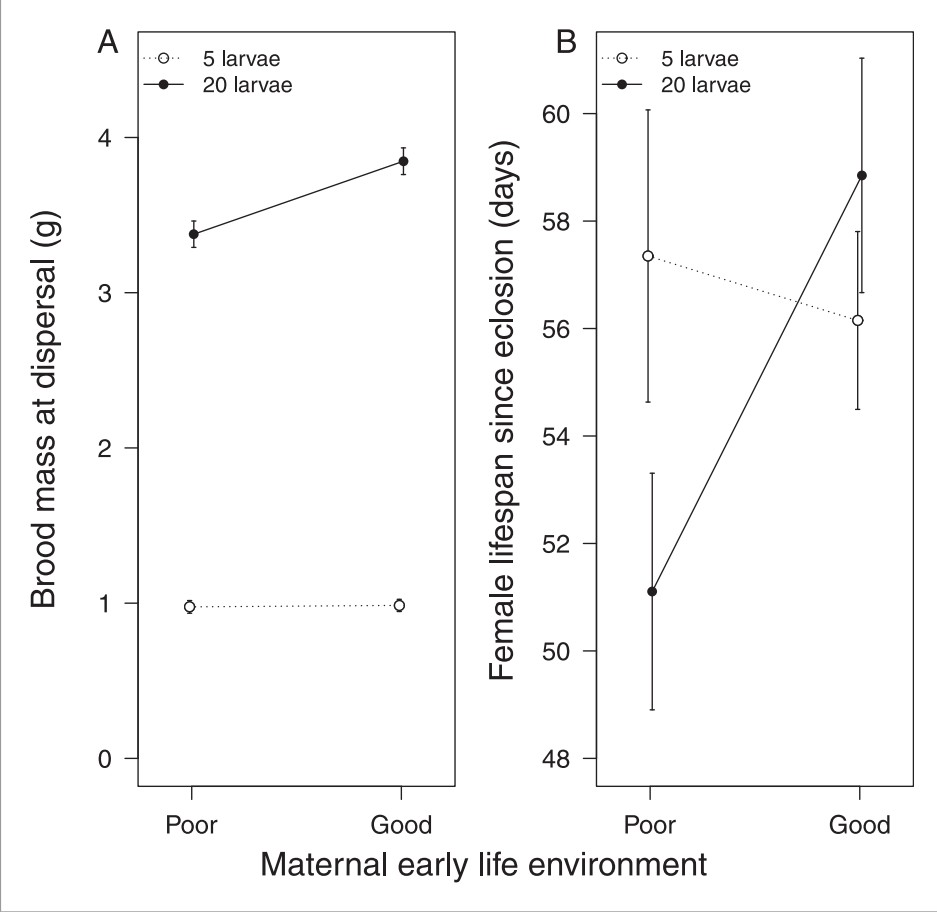

Figure 2. The effect of the mother's early-life environment on (**A**) brood mass at the dispersal stage of development and (**B**) her lifespan after reproduction. Mothers raised cross-fostered broods of either 5 (white circles) or 20 larvae (black circles), singlehandedly. A poor quality environment in early life generates mothers that (**A**) are less effective at raising broods of 20 cross-fostered larvae and (**B**) exhibit lower subsequent survival than mothers raised in a good quality early-life environment. Mean values with standard error bars are shown.

p = 0.41, *Figure 3A*). After raising broods of 20 larvae, fathers that received no post-hatching care also tended to have a shorter lifespan than fathers that were provisioned as larvae (*Table 1d*, *Figure 3B*).

## Experiment 2—the influence of parental effects on the outcome of a social interaction

The key finding of this experiment was that males suffered greater fitness costs when raising offspring with lower quality females ($X^2$ = 15.05, d.f. = 3, p = 0.002; *Figure 4*; *Table 2*). Males paired with females that had received no post-hatching care as larvae had significantly shorter lives than those whose partners received either 24-hr care (*Table 2b*) or 192-hr care as larvae (*Table 2b*). Independent of their partner's quality, males also had a shorter life if the mass of their brood was greater at dispersal (*Table 2b*) and if they were relatively small in size (*Table 2b*). By contrast, all the females in our experiment lived similarly long lives, irrespective of the conditions they experienced in early life ($X^2$ = 5.06, d.f. = 3, p = 0.17; *Figure 4*; *Table 2a*), or their size (*Table 2a*), or the total mass of their broods at dispersal (*Table 2a*).

This result cannot be attributed to males paired with low-quality females raising more offspring. Pairs where the female received no post-hatching care as a larva produced fewer offspring than those reared by females that received either 24-hr care (*Table 2c*) or 192 hr/full care (*Table 2c*).

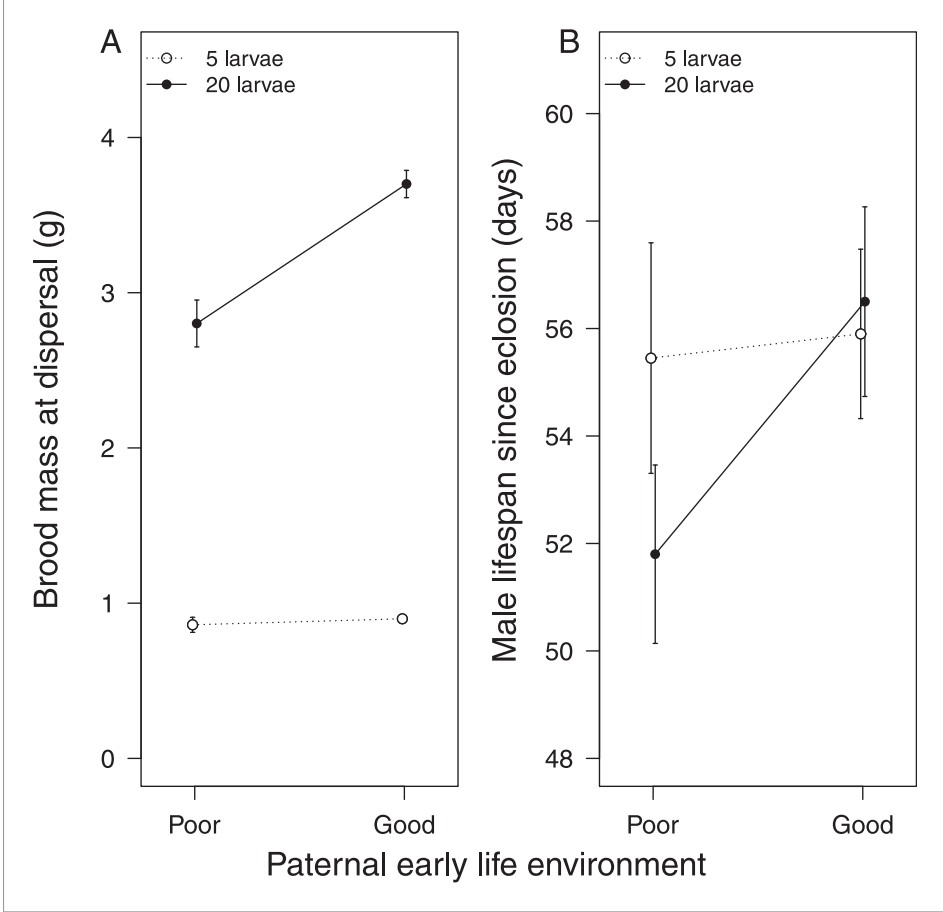

**Figure 3**. The effect of the father's early-life environment on (**A**) brood mass at the dispersal stage of development and (**B**) his lifespan after reproduction. Fathers raised cross-fostered broods of either 5 (white circles) or 20 larvae (black circles), singlehandedly. A poor quality environment in early life generates fathers that (**A**) are less effective at raising broods of 20 cross-fostered larvae and (**B**) tend to exhibit lower subsequent survival than fathers raised in a good quality early-life environment. Mean values with standard error bars are shown.

## Model: sexual conflict and the transgenerational impact of parental effects

Our model predicts that parents will provide more care when this increases the chances that their offspring develop into high-quality parents (*Figure 5*). Stronger parental effects selected for greater care regardless of whether sexual conflict was present or artificially removed from the model. When conflict was removed, however, parents both cared more in absolute terms and also increased their care more steeply in response to increasing parental effects (*Figure 5*). This is because sexual conflict dilutes the fitness benefits of investing in the current brood, whether these benefits arise from the quantity or the quality of offspring.

Good quality parents always invested more in care than poor quality parents in our model. The difference in care was large enough that good parents had higher overall mortality ($m_G > m_P$). The reproductive value of good parents was nonetheless always higher, in contrast to the predictions of a previous model (*Lessells and McNamara, 2012*). Poor quality individuals invested relatively less in care when partnered with good than with poor partners (i.e., $c_{PG} < c_{PP}$) and good quality individuals compensated for their partner's inferior efforts by increasing their own care ($c_{GP} > c_{GG}$). Interestingly, this was true both with and without sexual conflict (note that it is possible to remove conflict while retaining individual variation in parental quality, see Appendix 1). A reduction of care by individuals when paired with a high-quality partner consequently does not provide evidence for sexual conflict per se. However, when coparents are unrelated (as in the experiments and in the model with sexual conflict), a reduction in care by one individual will reduce the fitness of its partner and can consequently be considered exploitative.

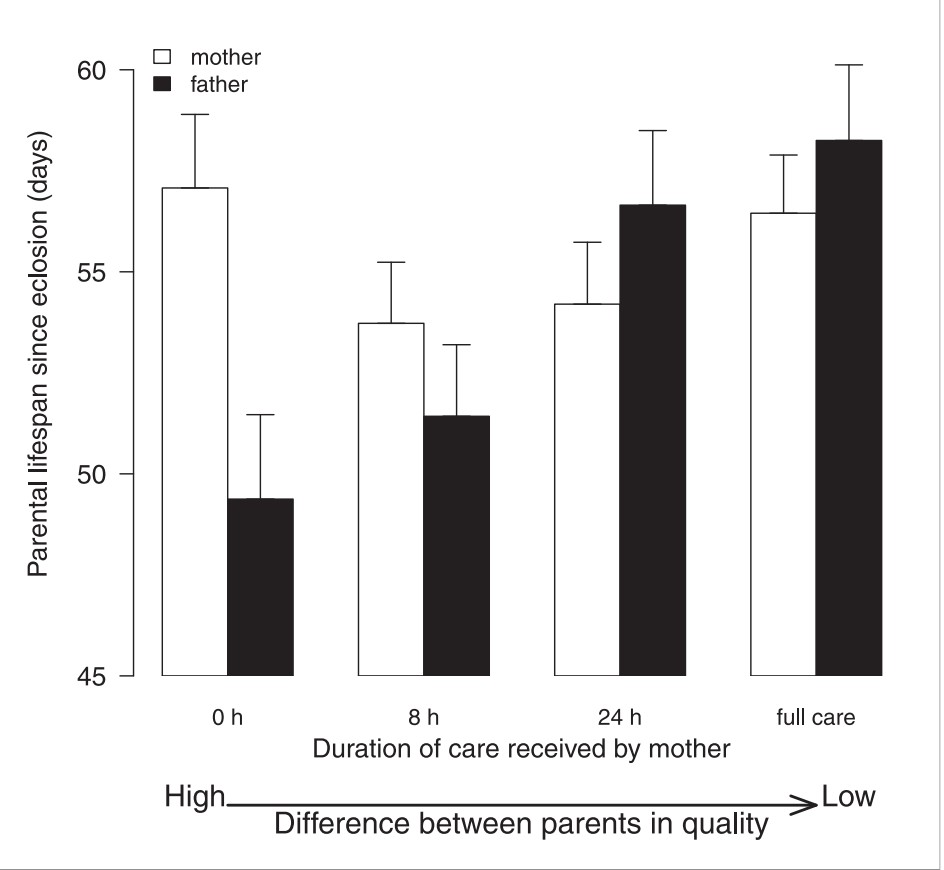

**Figure 4**. The effect of the female's early-life environment (i.e., the duration of care she received as a larva) on her lifespan after reproduction (white bars), and on the lifespan of the male with whom she raised offspring (black bars). All males developed in a high-quality environment. The greater the difference within the pair in the environment they each experienced during development, the greater the difference in their subsequent lifespan. Low-quality mothers thus exploit high-quality fathers. Mean values with standard error bars are shown.

## Discussion

It has long been appreciated that parents can influence the development of diverse behavioural traits in their young simply by varying aspects of the environment in which their offspring grow and develop (reviewed by e.g., *Champagne and Meaney, 2007*; *Daskalakis et al., 2013*; *Burton and Metcalfe, 2014*). Furthermore, it is well-understood that parents can change their offspring's fitness simply by influencing their offspring's size (e.g., *Steiger, 2013*). The work we present here provides further evidence in support of each of these well-established research findings. The novel contribution of our study is to identify three different ways in which parental effects can change the adaptive value of the offspring's behaviour in later life: by influencing the associated fitness benefits and costs, by changing the evolutionary outcome of social interactions, and by modifying the evolutionarily stable expression of behavioural traits that are themselves parental effects.

Our first experiments, using single parents, showed that the extent of parental care received during development affects the fitness costs and benefits associated with the supply of care, when larvae mature into parents themselves. Whether we examined mothers or fathers, we found that individuals that received no post-hatching care as larvae were less effective at raising a large brood as parents and sustained a greater associated fitness cost. By *Lessells and McNamara's (2012)* definition, these were low-quality parents, deriving relatively low fitness benefits (*Figures 2, 3*) from supplying care whilst paying a relatively high fitness cost (*Figures 2, 3*).

With our second experiment, using biparental care, we showed that these long-term developmental effects on parental quality can potentially change the outcome of social interactions

**Table 2**. Results from Experiment 2: the influence of parental effects on the outcome of a social interaction

| NB parental effect experienced by female | Coefficient | Standard error | z value | p value |
|---|---|---|---|---|
| a. Female lifespan | | | | |
| Parental effect: 8 hr vs 0 hr | 0.427 | 0.228 | 1.88 | 0.061 |
| Parental effect: 24 hr vs 0 hr | 0.216 | 0.236 | 0.92 | 0.360 |
| Parental effect: 192 hr vs 0 hr | −0.073 | 0.254 | −0.29 | 0.770 |
| Total carcass mass | 0.017 | 0.016 | 1.06 | 0.290 |
| Total brood mass | 0.004 | 0.029 | 0.16 | 0.870 |
| Female pronotum | 0.522 | 0.322 | 1.62 | 0.110 |

| NB parental effect experienced by male's partner | Coefficient | Standard error | z value | p value |
|---|---|---|---|---|
| b. Male lifespan | | | | |
| Parental effect: 8 hr vs 0 hr | −0.109 | 0.235 | −0.47 | 0.640 |
| Parental effect: 24 hr vs 0 hr | −0.664 | 0.252 | −2.63 | 0.008 |
| Parental effect: 192 hr vs 0 hr | −1.033 | 0.291 | −3.55 | 0.0003 |
| Total carcass mass | −0.027 | 0.023 | −1.17 | 0.240 |
| Total brood mass | −0.094 | 0.032 | −2.87 | 0.004 |
| Male pronotum | 0.915 | 0.366 | 2.50 | 0.012 |

| NB parental effect experienced by brood's mother | Estimate | Standard error | z value | p value |
|---|---|---|---|---|
| c. Brood size | | | | |
| Intercept | 1.35776 | 0.39286 | 3.456 | 0.0005 |
| Parental effect: 8 hr vs 0 hr | 0.02350 | 0.03295 | 0.713 | 0.476 |
| Parental effect: 24 hr vs 0 hr | 0.16261 | 0.03453 | 4.710 | <0.0001 |
| Parental effect: 192 hr vs 0 hr | 0.14641 | 0.03676 | 3.983 | <0.0001 |
| Total carcass mass | 0.00765 | 0.00308 | 2.485 | 0.013 |
| Female pronotum | 0.31190 | 0.05133 | 6.076 | <0.0001 |
| Male pronotum | 0.12973 | 0.05729 | 2.265 | 0.024 |

Parental effects were created experimentally by exposing females to 0 hr, 8 hr, 24 hr, or 192 hr of post-hatching care as larvae. They were then kept until adulthood and allowed to breed twice with a male who had received 192 hr of care as larva. The two parents raised offspring together. Each parent's lifespan thereafter was recorded, as was the mass of their brood at dispersal. Further details are given in the 'Materials methods'.

during biparental care in adulthood. Just as predicted by theory (*Lessells and McNamara, 2012*), we found that the costs of care were divided most unevenly between the sexes when the difference within the pair in parental quality was at its greatest (*Figure 4*). High-quality males had lower residual fitness after raising offspring with low-quality females than their brothers did after raising young with high-quality female partners, losing up to 8 days from their lifespan. Under these conditions, and according to the fitness-based definition of selfish exploitation we are using here (see 'Introduction'), males were exploited by their lower quality partners. Based on previous behavioural observations of biparental care in *N. vespilloides*, the most likely explanation is that males with low-quality partners put more effort into parental duties to compensate for the shortcomings of their mate and paid a higher fitness cost accordingly (*Walling et al., 2008*; *Ward et al., 2009*; *Cotter et al., 2010*). How these differences in lifespan translate into reproductive success in nature is unknown, but since males might mate with multiple females in 8 days (*Eggert, 1992*), the effects are unlikely to be trival.

The biparental care experiment yielded an unexpected result. Although lower quality females paid a higher cost for raising offspring as single mothers (*Figure 2*), all females paid a similar cost under

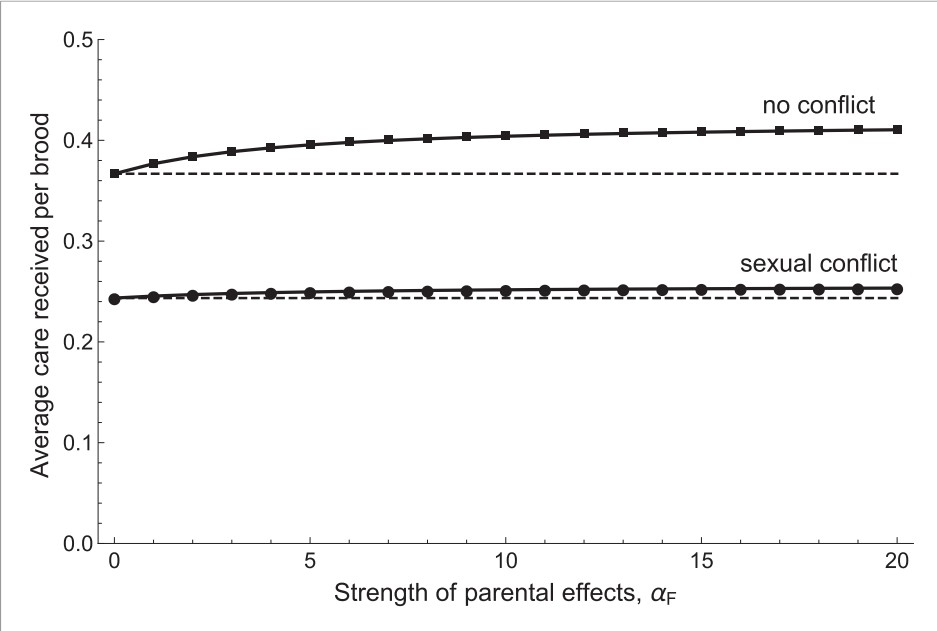

**Figure 5**. The relationship between the average care received by a brood and the strength of parental effects $\alpha_F$ (i.e., the extent to which care received affects the future parental quality of offspring). Average care levels increase with the strength of parental effects both when sexual conflict is present (circles) and when it is absent (squares). When there is no sexual conflict, parents provide more care in absolute terms and also increase their care more steeply with increasing parental effects. Shown with $g_1 = 0.5$, $b_{min} = f_{min} = 1$, $\alpha_B = 5$, $\beta_G = 5$, $\beta_P = 5$, and $m_{min} = 0.25$ (see Appendix 1 for details of function shapes).

biparental care, irrespective of their quality (**Figure 4**). We suggest two explanations for these findings. One possibility is that they result from slight methodological differences between our first and second experiments. In the first experiment, we created standard broods of 5 and 20 larvae for our experimental subjects to raise, whereas in the second experiment, parents raised a brood size of their choosing. The former approach may have been more effective at exposing costs of care than the latter (see **Lessells, 1991**). An alternative possibility is that the results reveal strategic differences in the rules for raising offspring that depend on whether females raise offspring alone or with a partner. Perhaps when raising offspring with a partner, the female's contributions to care are calibrated more in relation to the costs associated with care, than in relation to the benefits gained. Conversely, when rearing young alone, maternal investment is perhaps determined more by the benefits to be gained rather than the costs that are borne. Which of these explanations better accounts for the results remains to be determined with further experiments.

A key discovery of the biparental care experiment is that it reveals a novel potential social cost associated with receiving high levels of care during development, which is not borne until adulthood when individuals engage in biparental care themselves. The cost arises when a high-quality parent is paired with a low-quality partner and it is apparently caused by the lower quality parent forcing the higher quality parent to bear a greater share of the costs of parental investment.

These findings raise two further questions. First, how likely is it that this social cost will ever show itself in nature, in any species that exhibits biparental care? Many species with biparental care also exhibit assortative pairing (e.g., **Jiang et al., 2013**), and this will prevent the sort of mismatches in quality that we generated experimentally, and hence reduce the potential for exploitation of higher quality parents through sexual conflict. Our experimental results thus suggest a novel function of assortative pairing: to minimize differences in partner quality, and so reduce the social cost imposed by sexual conflict that we have uncovered here.

In burying beetles, though, assortative pairing by mate choice is unlikely to be the norm because there is little evidence that individuals ever reject a potential mate (**Scott, 1998**). It might be argued that assortative pairing could instead arise as a by-product of fights for carcass ownership, because

there can be intense competition within each sex over a carcass, which the largest individuals usually win (e.g., *Hopwood et al., 2013*). The dominant individuals of each sex then pair up and prepare the carcass together. However, it is unclear how frequently carcass ownership is contested in natural populations. In nature, adult burying beetles fly to locate carrion, which is a key resource for reproduction (*Scott, 1998*), often covering long distances in their search (*Attisano and Kilner, 2015*). Therefore, individuals are distributed patchily and at relatively low densities, according to the distribution of carrion. A field study in which carrion was put out for burying beetles found that ownership of the carcass was contested within both sexes in 22 out of 42 breeding events (*Müller et al., 2007*). Whether the dominant pair were closely matched in body size, and more closely matched than pairs at uncontested carcasses, is not known. One interpretation of these different strands of evidence is that assortative pairing by quality is unlikely at least half of the time in *N. vespilloides*, and on these occasions therefore, high-quality burying beetles parents are vulnerable to exploitation by inferior quality partners. However, it is hard to draw strong inferences from the field study because it probably increased the size of the local burying beetle population by drawing individuals in to carcasses presented at a relatively high density, and so might have changed the extent of competition for a carcass.

The second question raised by our discovery of this social cost in the next generation is how does it feed back to influence evolutionarily stable levels of parental care in the current generation? Previous theoretical work has focused on contemporary negotiations between partners to deduce the best strategy of care and has rather neglected transgenerational costs of the sort we have identified here. Therefore to address this question, we needed to develop new theory. Our theoretical work showed that the threat of exploitation during biparental care in the next generation changes current interactions between parents and their young. It limits the potential benefits associated with the provision of care and so reduces the evolutionarily stable level of care currently supplied to offspring (*Figure 5*). Thus, our model identifies a third way in which parental effects can influence the adaptive value of behavioural traits. It shows that when behavioural traits are themselves parental effects, and can change the adaptive value of a behavioural trait in the next generation, then evolutionarily stable levels of behaviour in the current generation will change accordingly as well.

Could these transgenerational effects still persist, even under assortative mating? Although our model assumes random mating with respect to parental quality, we expect that our results would remain qualitatively unchanged under assortative mating. Assortative mating would increase the benefits of investing in good quality parents-to-be, because good parents would on average find better partners. We would consequently expect the average level of care provided to be higher when there is assortative mating, rather than random mating, and to increase more steeply with the strength of parental effects. Nonetheless, even when mating is highly assortative, there is still sexual conflict over the level of care provided by each parent, and this should reduce the total care provided.

The evolution of adaptive behaviour depends both on a mechanism for inheritance of behavioural traits from generation to generation and on the net fitness gained by performing that behaviour. We have shown here that parental effects not only influence the inheritance of a behavioural trait (because well-cared for larvae mature into high-quality parents) but also alter the fitness associated with performing that behaviour—and in three contrasting, yet interconnected ways. Thus, our experiments show how parental effects can potentially provide a mechanism for the rapid evolution of new adaptive social behaviour in the face of environmental change (see also *Badyaev and Uller, 2009*). Whether parental effects similarly change the fitness costs and benefits associated with other social behaviours that might also function as parental effects, such as cooperative breeding, an individual's position in a social network, or even interspecific mutualisms, remains an exciting challenge for future work.

## Acknowledgements

We are very grateful to A Backhouse for maintaining the burying beetle colony and to K McGhee, N Boogert, M Schrader, I Baldwin, T Szekely, and C Bergstrom for their comments on early drafts of the manuscript. RMK was supported by a Wolfson Merit Award from the Royal Society; RMK and BJMJ were funded by an ERC Consolidators grant 310785 Baldwinian_Beetles to RMK; GB was funded by Marie Curie Intra-European Fellowship 252120 LIFHISBURBEE; JMH was funded by an Australian Postgraduate Award; O De G was funded CONACYT and by the Cambridge Trust.

# Additional information

## Funding

| Funder | Grant reference | Author |
|---|---|---|
| European Research Council (ERC) | 310785 -BALDWINIAN_BEETLES | Rebecca M Kilner |
| Consejo Nacional de Ciencia y Tecnología | Graduate Student Fellowship | Ornela De Gasperin |
| Cambridge Trust | Graduate Student Fellowship | Ornela De Gasperin |
| Marie Curie Intra-European Fellowship | PIEF-GA-2009-252120 | Giuseppe Boncoraglio |
| Australian Postgraduate Award | Graduate Student Fellowship | Jonathan M Henshaw |

The funders had no role in study design, data collection and interpretation, or the decision to submit the work for publication.

## Author contributions

RMK, HK, Conception and design, Analysis and interpretation of data, Drafting or revising the article; GB, Conception and design, Acquisition of data, Analysis and interpretation of data; JMH, Conception and design, Acquisition of data, Analysis and interpretation of data, Drafting or revising the article; BJMJ, Analysis and interpretation of data, Drafting or revising the article; ODG, AA, Acquisition of data, Analysis and interpretation of data

## Author ORCIDs

Jonathan M Henshaw, http://orcid.org/0000-0001-7306-170X

# Additional files

## Major dataset

The following dataset was generated:

| Author(s) | Year | Dataset title | Dataset ID and/or URL | Database, license, and accessibility information |
|---|---|---|---|---|
| Kilner RM, Boncoraglio G, Henshaw JM, Jarrett BJM, De Gasperin O, Kokko H | 2015 | Parental effects alter the evolutionary economics of social interactions within the family | http://dx.doi.org/10.5061/dryad.fh34h | Available at Dryad Digital Repository under a CC0 Public Domain Dedication. |

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

## Appendix 1

### Reproductive value of mutants that differ in their behaviour

In the main text, we calculated the reproductive value of typical individuals in the population. Here, we consider the reproductive value of rare mutant individuals that differ in their care behaviour from the population norm. Since these mutants are rare, we assume that they always mate with typical individuals. Half of the resulting offspring are mutants and half are typical. These simplifying assumptions are only needed to calculate the direction of selection on care levels, so we do not expect them to affect our conclusions.

The equations in this section are completely analogous to those in the main text. We mark any variable or parameter that relates to a mutant individual with a hat. For instance, we write $\widehat{c}_{AB}$ for the amount of care provided by a mutant of type $A$ whose partner is of type $B$.

First, the mortality risk of a mutant of type $A$ after mating with a typical individual of type $B$ is

$$\widehat{m}_{AB} = m_A(\widehat{c}_{AB}). \tag{21}$$

The average mortality risk between reproductive bouts for good and poor mutants is then, respectively,

$$\widehat{m}_G = g_2\widehat{m}_{GG} + (1 - g_2)\widehat{m}_{GP}, \tag{22}$$

and

$$\widehat{m}_P = g_2\widehat{m}_{PG} + (1 - g_2)\widehat{m}_{PP}. \tag{23}$$

Next, the expected brood productivity from a mating between a mutant of type $A$ and a typical individual of type $B$ is

$$\widehat{b}_{AB} = b(\widehat{c}_{AB} + c_{BA}). \tag{24}$$

Note that, unlike their analogues for matings between typical individuals, $\widehat{b}_{GP}$ and $\widehat{b}_{PG}$ may not be equal. We normalize the expected number of surviving offspring as

$$\widehat{b}'_{AB} = \frac{\widehat{b}_{AB}}{\overline{b}}, \tag{25}$$

where $\overline{b}$ is the average lifetime reproductive success in the population (see **Equation 6** in the main text). The relative competitiveness of offspring from this mating is similarly

$$\widehat{f}_{AB} = b(\widehat{c}_{AB} + c_{BA}). \tag{26}$$

The probability $\widehat{g}_{AB}$ that any particular offspring develops into a good parent is then given by solving

$$(1 - \widehat{g}_{AB})^{1/\widehat{f}_{AB}} = (1 - \widehat{g}_{GG})^{1/f_{GG}}. \tag{27}$$

We can now calculate the reproductive value of good and poor quality mutants. Remembering that the offspring of a mutant and a typical individual inherit each parent's allele with a probability of one half, these are given by

$$\widehat{v}_G = \frac{1}{2m_G}\left\{\left[g_2\widehat{b}'_{GG}\widehat{g}_{GG} + (1 - g_2)\widehat{b}'_{GP}\widehat{g}_{GP}\right]\left(\frac{v_G + \widehat{v}_G}{2}\right)\right.$$
$$\left. + \left[g_2\widehat{b}'_{GG}(1 - \widehat{g}_{GG}) + (1 - g_2)\widehat{b}'_{GP}(1 - \widehat{g}_{GP})\right]\left(\frac{v_P + \widehat{v}_P}{2}\right)\right\}, \tag{28}$$

and

$$\widehat{v}_P = \frac{1}{2m_P} \left\{ \left[ g_2 \widehat{b}'_{PG} \widehat{g}_{PG} + (1-g_2) \widehat{b}'_{PP} \widehat{g}_{PP} \right] \left( \frac{v_G + \widehat{v}_G}{2} \right) + \left[ g_2 \widehat{b}'_{PG} (1 - \widehat{g}_{PG}) \right. \right.$$
$$\left. \left. + (1-g_2) \widehat{b}'_{PP} (1 - \widehat{g}_{PP}) \right] \left( \frac{v_P + \widehat{v}_P}{2} \right) \right\}. \tag{29}$$

These equations can be solved simultaneously for $\widehat{v}_G$ and $\widehat{v}_P$ using the values for $v_G$ and $v_P$ from **Equation 17** in the main text.

## Removing sexual conflict

In order to assess the importance of sexual conflict over parental care, we constructed a parallel model in which coparents are genetically identical (i.e., $r = 1$ between coparents). Mutants invade in genetically uniform groups, with mutant offspring returning to their parent group, and we measure the fitness of a mutant group by its growth rate when it is small relative to the population of typical individuals. While biologically somewhat artificial, this scenario removes sexual conflict entirely while retaining all other features of the original model (note in particular that the mutants within a group can differ in parental quality despite being genetically identical, as they may have experienced different early-life conditions; but differences in behaviour based on phenotype are now more analogous to arms and legs of a single individual human acting differently to aid in the joint task of efficient locomotion, than to a conflict scenario where one part tries to make the other do more work).

All equations applying to typical individuals carry over without change from the original model, but the equations applying to mutants often differ. As above, the mortality risk of a mutant of quality $A$ after mating with a partner of quality $B$ is given by

$$\widehat{m}_{AB} = m_A(\widehat{c}_{AB}). \tag{30}$$

Since mutants always mate with other mutants, the brood productivity of a mutant pair of qualities $A$ and $B$ is

$$b_{AB} = b(\widehat{c}_{AB} + \widehat{c}_{BA}). \tag{31}$$

As in the original model, we normalize this to

$$\widehat{b}'_{AB} = \frac{\widehat{b}_{AB}}{\overline{b}}, \tag{32}$$

where $\overline{b}$ is the average lifetime reproductive success in the population (see **Equation 6** in the main text). The relative competitiveness of offspring from this pair is

$$\widehat{f}_{AB} = f(\widehat{c}_{AB} + \widehat{c}_{BA}). \tag{33}$$

The probability $\widehat{g}_{AB}$ that any particular offspring develops into a good parent is then given by solving

$$(1 - \widehat{g}_{AB})^{1/\widehat{f}_{AB}} = (1 - g_{GG})^{1/f_{GG}}. \tag{34}$$

Since mutant offspring always return to their parents' group, the proportion of good parents among newly recruited individuals $\widehat{g}_1$ is no longer fixed, rather depending on how much care mutants provide relative to typical individuals. To find its value, we divide the average number of good parents produced by a mutant over its lifetime by the total average brood productivity of mutants:

$$\widehat{g}_1 = \frac{\left(\frac{\widehat{g}_1}{\widehat{m}_G}\right)\left(\widehat{g}_2\widehat{b}_{GG}\widehat{g}_{GG} + (1-\widehat{g}_2)\widehat{b}_{GP}\widehat{g}_{GP}\right) + \left(\frac{1-\widehat{g}_1}{\widehat{m}_G}\right)\left(\widehat{g}_2\widehat{b}_{PG}\widehat{g}_{PG} + (1-\widehat{g}_2)\widehat{b}_{PP}\widehat{g}_{PP}\right)}{\left(\frac{\widehat{g}_1}{\widehat{m}_G}\right)\left(\widehat{g}_2\widehat{b}_{GG} + (1-\widehat{g}_2)\widehat{b}_{GP}\right) + \left(\frac{1-\widehat{g}_1}{\widehat{m}_G}\right)\left(\widehat{g}_2\widehat{b}_{PG} + (1-\widehat{g}_2)\widehat{b}_{PP}\right).} \tag{35}$$

The proportion of good parents among mating adults $\widehat{g}_2$ is then given by

$$\frac{\widehat{g}_2 = \widehat{g}_1/\widehat{m}_G}{\widehat{g}_1/\widehat{m}_G + (1-\widehat{g}_1)/\widehat{m}_P.} \tag{36}$$

Further, the overall mortality risks $\widehat{m}_G$ and $\widehat{m}_P$ for good and poor parents are

$$\widehat{m}_G = \widehat{g}_2\widehat{m}_{GG} + (1-\widehat{g}_2)\widehat{m}_{GP}, \tag{37}$$

$$\widehat{m}_P = \widehat{g}_2\widehat{m}_{PG} + (1-\widehat{g}_2)\widehat{m}_{PP}. \tag{38}$$

We can obtain values for $\widehat{g}_1$, $\widehat{g}_2$, $\widehat{m}_G$, and $\widehat{m}_P$ by solving **Equation 34** through **Equation 38** simultaneously using numerical methods.

Since mutants do not mate with typical individuals, we define selection gradients on the initial rate of increase of mutants in the population, rather than on reproductive values. This method gives identical results if applied to the original model. Suppose we write $n_G$ and $n_P$ for the numbers of good and poor quality mutants in one generation, and $n'_G$ and $n'_P$ for their numbers in the next. By similar arguments to those in the main text, we have

$$\begin{bmatrix} n'_G \\ n'_P \end{bmatrix} = \widehat{M} \begin{bmatrix} n_G \\ n_P \end{bmatrix}, \tag{39}$$

where the entries of $\widehat{M}$ are given by

$$\widehat{M}_{11} = \frac{1}{\widehat{m}_G}\left[\widehat{g}_2\widehat{b}'_{GG}\widehat{g}_{GG} + (1-\widehat{g}_2)\widehat{b}'_{GP}\widehat{g}_{GP}\right]$$

$$\widehat{M}_{12} = \frac{1}{\widehat{m}_P}\left[\widehat{g}_2\widehat{b}'_{GP}\widehat{g}_{GP} + (1-\widehat{g}_2)\widehat{b}'_{PP}\widehat{g}_{PP}\right]$$

$$\widehat{M}_{21} = \frac{1}{\widehat{m}_P}\left[\widehat{g}_2\widehat{b}'_{GG}(1-\widehat{g}_{GG}) + (1-\widehat{g}_2)\widehat{b}'_{GP}(1-\widehat{g}_{GP})\right]$$

$$\widehat{M}_{22} = \frac{1}{m_P}\left[\widehat{g}_2\widehat{b}'_{GP}(1-\widehat{g}_{GP}) + (1-\widehat{g}_2)\widehat{b}'_{PP}(1-\widehat{g}_{PP})\right]. \tag{40}$$

The maximum initial growth rate of the mutant group is given by the principal eigenvalue of $\widehat{M}$, which we denote $\lambda$. We can then define a selection gradient on the levels of care $c = (c_{GG}, c_{GP}, c_{PG}, c_{PP})$ as

$$S(c) = \left(\frac{\partial\lambda}{\partial\widehat{c}_{GG}}, \frac{\partial\lambda}{\partial\widehat{c}_{GP}}, \frac{\partial\lambda}{\partial\widehat{c}_{PG}}, \frac{\partial\lambda}{\partial\widehat{c}_{PP}}\right)\Big|_{\widehat{c}=c}. \tag{41}$$

We are only able to calculate $\lambda(\widehat{c})$ numerically, rather than as an explicit function of the mutant care levels $\widehat{c}$. Consequently, we approximate each component $i$ of the selection gradient as

$$S_i \approx \frac{\lambda(c + \delta \cdot e_i) - \lambda(c - \delta \cdot e_i)}{2\delta}, \tag{42}$$

where $\delta$ is a small positive constant (we used $\delta = 10^{-3}$) and $e_i$ is the $i$th unit vector. As in the original model, we then locate evolutionarily stable levels of care by following the selection trajectories defined by $S$. We start with arbitrary initial levels of care $c = c_0$ and then iterate the equation

$$c_{t+1} = c_t + \Delta S(c_t). \tag{43}$$

We again found $\Delta = 0.01$ suitable.

## Choice of functions for mortality, brood productivity, and transgenerational effects

All equations above and in the main text hold for any choices of the functions that relate parental care to offspring survival, offspring competitiveness, and adult mortality. In order to run the model, however, we must specify each of these functions explicitly. We assume that brood productivity and offspring competitiveness are increasing concave functions of the total amount of care received $c$ and that mortality is an increasing concave function of $c$. This ensures that populations always evolve iteroparity with some level of parental care.

In example runs of the model (**Figure 5**), we assumed that brood productivity and offspring competitiveness are given by

$$b(c) = b_{min} + \frac{\alpha_B c}{1+c}, \tag{44}$$

and

$$f(c) = f_{min} + \frac{\alpha_F c}{1+c}. \tag{45}$$

Here, $b_{min}$ and $f_{min}$ are the brood productivities and competitiveness of larvae that receive no parental care. Steeper values of $\alpha_B$ and $\alpha_F$ correspond to greater benefits of increased care.

We assumed that the mortality risk for good and poor parents after providing parental care of is given, respectively, by

$$m_G(c) = m_{min} + \beta_G c^2, \tag{46}$$

and

$$m_P(c) = m_{min} + \beta_P c^2. \tag{47}$$

Here, $0 < m_{min} < 1$ is the mortality risk of an individual that provided no parental care in the previous reproductive bout. The constants $\beta_G$ and $\beta_P$ determine how quickly mortality increases with additional parental care. We ensure that poor parents have higher mortality than good parents for any given level of care by assuming that $\beta_P > \beta_G > 0$. The maximum possible levels of care are $\sqrt{(1-m_{min})/\beta_G}$ and $\sqrt{(1-m_{min})/\beta_P}$ for good and poor parents, respectively.

