## [Decision Letter]

Thank you for submitting your work entitled “Parental effects alter the evolutionary economics of social interactions within the family” for peer review at *eLife*. Your submission has been favorably evaluated by Ian Baldwin (Senior Editor), a Reviewing Editor, and two reviewers.

The following individuals responsible for the peer review of your submission have agreed to reveal their identity: Carl Bergstrom (Reviewing Editor) and Tamás Székely (peer reviewer).

The reviewers have discussed the reviews with one another and the Reviewing editor has drafted this decision to help you prepare a revised submission.

The authors present an interesting study on burying beetles that couples experimental work on parental effects and parent care costs to a theoretical model in which sexual conflict reduces the incentives for high parental investment in offspring quality. Thus, the authors argue, parental care in one generation shapes the social interaction of parental care in the next generation. This is an interesting result and potentially could merit publication in *eLife*. However, major revision will be necessary.

Essential revisions:

1) Isn't “quality” as resulting from parental care environment just a vague surrogate for body size? If so, the authors should be clear that these are size-mediated effects and thus they could arise from many other environmental effects other than extent of parental care. Is there any way to differentiate the effects of size and some other measure of quality? In nature, do males or females prefer larger mates? Competition for carcasses should favor larger beetles; so maybe in the natural system individuals seldom encounter or choose small individuals.

2) In the experiments, matings are forced and there is no option for intrasex competition or mate choice. The authors minimize this by noting that population density is low and there is no documented direct mate choice in this species. Yet there is a high probability that mate choice occurs as a result of competition for carcasses where larger individuals almost always win. So, only in low density systems would you expect individuals to mate with any beetle of opposite sex on the carcass. Moreover if a male mates with a low-quality female and then leaves the carcass he can completely avoid the potential negative effect of exploitation. The model also precludes this sort of mate selection arising from competition. The authors need to address this issue and consider how it may influence their results.

3) The referees all felt that the organization of the paper and the presentation of the Results were not optimal. The Methods should precede the Results; this will also reduce the current redundancy between these two sections. The actual results are presented in an overly casual form: the authors need to present full statistical results including ANOVA tables or their equivalent, followed by some explanation of effect size. Interpretation of the results should be deferred to the Discussion section rather than presented in the Results section.

I think the data should be re-evaluated in the context of a size effect and the hypothesis of negative social effects arising from parental care environment should be dropped.

4) One referee notes that while there is relatively little work on the effects of early life experience on parenting, it would worth digging in the ethological literature. In particular, are there relevant studies in the work of Klaus Immelmann and Bielefeld and followers? In numerous experiments they've manipulated early life experiences in zebra finches and looked to see the implications of these experiences in adulthood. In addition, what about Michael Meaney's work about how good mothers produce less stressed daughters in rats and the Muller et al. (2011 Auk) study about how abused chicks became abusing parents themselves in boobies?

5) A fundamental difference between experiment 1 and experiment 2 seems to be that in the former a set of young (20 or 5) was given to the parents, whereas in the latter the females produced clutch size they wanted. Could this difference affect why female survival was influenced in experiment 1 but not experiment 2? What drives the difference in the number of offspring produced in experiment 2: different number of eggs produced, or differential provisioning of parental care?

6) The model seems rather divorced from the rest of the paper, and under-emphasized as well. The model should be better integrated with the main flow of the paper. Perhaps an additional figure is needed to illustrate the model schematically.

7) Figure 5 does not yet provide a compelling illustration of the sexual conflict result. From this figure it is immediately clear whether the slower rate of increase in biparental parental care with α_F_ results from the sexual conflict story rather than simply as a result of lower marginal returns on investment given the higher baseline investment level given biparental care.

8) The authors need to think carefully about what it means to claim that females exploit high quality males. The authors show in Figure 4 that males suffer from having lower-quality mates. But this is not necessarily exploitation; these lower quality mates could be doing the best they are able and the result would still obtain. I would think that at the bare minimum, exploitation would involve females contributing lower levels of parental care when paired with higher quality males. To be clear, I am not asking for additional experiments to test this, just for a reconsideration of what one must observe to demonstrate exploitation.

[Editors' note: further revisions were requested prior to acceptance, as described below.]

Thank you for resubmitting your work entitled “Parental effects alter the adaptive value of an adult behavioural trait” for further consideration at *eLife*. Your revised article has been favorably evaluated by Ian Baldwin (Senior Editor), Carl Bergstrom (Reviewing Editor), and one of the original reviewers. We appreciate the diligence with which you have addressed the reviewers' concerns, and feel that the manuscript has been improved but there are some remaining minor issues that need to be addressed before acceptance, as outlined below:

From Reviewer 1:

The authors have done a good job of differentiating quality from body size. Both effects are included in the analysis and the results are consistent in addition to the size effect. I do think the authors should address the size effect a little more directly. I suggest treating this in the Discussion to make clear to readers that the quality effect is in addition to size effects.

Lifespan as a measure of fitness is OK, but the effect size shown here (i.e., 4–8 days of additional life) seems quite modest. I agree that they have demonstrated that potential exploitation of high quality individuals is a risk, but the risk seems low and the cost in terms of lifespan seems low as well compared to the potential benefits of increased brood size and quality.

In their effort to determine if this sort of pairing (low quality with high quality) happens in nature, they paraphrase a study where carcasses were added to the environment. Half of the carcasses were uncontested in the study, and the authors suggest this provides a conservative estimate. However, by increasing the number of carcasses, the cited study actually causes an effective decrease in density of beetles. Because carcasses are likely less abundant than in this study, most carcasses are likely to be contested under natural conditions. Thus, the authors may be overestimating the likelihood of uncontested carcasses and the coincident possibility of exploitation by low quality individuals.

From the Reviewing Editor:

I suggest that you expand the Abstract to better describe the specifics of what you have found in the experiments: (1) receiving high parental care as a juvenile results in individuals having higher success at raising large broods and lower mortality from doing so; (2) high quality males have to compensate for their low-quality partners and thus suffer higher mortality rates than high quality males with high quality partners.

In the subsection “Outline of model”, please stress that the “sealed bid” in the model is (if I understand correctly) contingent on the type of the partner, not a single bid irrespective of partner type.

In the Discussion section, the authors state: “Under these conditions, and according to the fitness-based definition of selfish exploitation we are using here (see Introduction), males were exploited by their lower quality partners”. But in the model, I think this happens even when mates are genetically identical. Would you say that they are “exploiting” one another? It is hard to envision why selection would favor true exploitation in that case. Please clarify.

---

## [Author Response]

Essential revisions:

1) Isn't “quality” as resulting from parental care environment just a vague surrogate for body size?

No. We use a precise definition of ‘quality’, which was originally set out in the theoretical paper by Lessells and McNanama (2012). Although we explained this in the original version of the manuscript, it was evidently easy to miss. Therefore we have spelled this out much more explicitly in the revised version (Introduction).

If so, the authors should be clear that these are size-mediated effects and thus they could arise from many other environmental effects other than extent of parental care. Is there any way to differentiate the effects of size and some other measure of quality?

Our definition of quality is based on measures of fitness: fitness gained through parental care as well as fitness concurrently lost. A high quality individual gains high fitness from parental care for a relatively low loss of fitness. Size is obviously a completely different measure, and not one that we are interested in here.

In nature, do males or females prefer larger mates? Competition for carcasses should favor larger beetles; so maybe in the natural system individuals seldom encounter or choose small individuals.

We have added a section in the Discussion explaining that there is no evidence for direct mate choice in burying beetles. We also explain how size might influence competition for a carcass and explain in the Methods that the range in size of our experimental beetles closely matches the range we see in field caught beetles (subsection “Generation 1—manipulation of larval developmental environment via a parental effect” and Figure 1—figure supplement 2). From all the available evidence, small individuals are encountered in the natural system.

2) In the experiments, matings are forced and there is no option for intrasex competition or mate choice. The authors minimize this by noting that population density is low and there is no documented direct mate choice in this species. Yet there is a high probability that mate choice occurs as a result of competition for carcasses where larger individuals almost always win.

This is a very reasonable point, so we went back to the literature to see if was possible to determine the probability of this happening in nature. The only data we could find for *N. vespilloides* come from an experiment in which mice were placed at artificially high densities in the field to attract burying beetles (Muller et al., 2007). This experiment found that at slightly over half (22/42) of all carcasses, ownership was contested by members of each sex. So not the high probability the referee suggests, and anyway probably higher than might be seen at natural densities of carcasses. We describe this study in the Discussion.

So, only in low density systems would you expect individuals to mate with any beetle of opposite sex on the carcass.

Yes – and, from the data above, these are evidently more common than the referee supposes.

Moreover if a male mates with a low-quality female and then leaves the carcass he can completely avoid the potential negative effect of exploitation.

True – and this may explain in part why males leave the carcass earlier than females (e.g. Boncorgalio et al., 2012). But by leaving immediately, the male also runs the risk that there will be an infanticidal takeover of the carcass because he has not stayed to defend it (e.g. [42]). The cost of losing all his reproductive success on the carcass probably outweighs the cost of experiencing a slightly reduced residual fitness, which may be why the male does not leave straightaway.

The model also precludes this sort of mate selection arising from competition. The authors need to address this issue and consider how it may influence their results.

The main results of our model are that the level of care provided to offspring increases with the strength of parental effects, but that it is reduced by sexual conflict between coparents. We expect that these results would still hold if mating were assortative. Assortative mating would increase the benefits of investing in good quality parents-to-be, because good parents would on average find better partners. We would consequently expect the average level of care provided to be higher under assortative than random mating. Nonetheless, even if mating is highly assortative, there is still sexual conflict over the level of care provided by each parent, and this should reduce the total care provided. We now address this point in the Discussion.

3) The referees all felt that the organization of the paper and the presentation of the Results were not optimal. The Methods should precede the Results; this will also reduce the current redundancy between these two sections. The actual results are presented in an overly casual form: the authors need to present full statistical results including ANOVA tables or their equivalent, followed by some explanation of effect size. Interpretation of the results should be deferred to the Discussion section rather than presented in the Results section.

We have changed the organization of the paper, and tightened the focus. The Results are now presented in tables and discussion has been removed from the Results.

I think the data should be re-evaluated in the context of a size effect and the hypothesis of negative social effects arising from parental care environment should be dropped.

As we explain above, there is no ‘size effect’. Instead our experiments show a direct effect of the early life environment on the fitness benefits and costs associated with supplying parental care in later life. Therefore we think our hypothesis of possible negative social effects arising from the parental care environment is valid, albeit with some caveats which we now spell out more explicitly in the Discussion.

4) One referee notes that while there is relatively little work on the effects of early life experience on parenting, it would worth digging in the ethological literature. In particular, are there relevant studies in the work of Klaus Immelmann and Bielefeld and followers? In numerous experiments they've manipulated early life experiences in zebra finches and looked to see the implications of these experiences in adulthood. In addition, what about Michael Meaney's work about how good mothers produce less stressed daughters in rats and the Muller et al. (2011 Auk) study about how abused chicks became abusing parents themselves in boobies?

We now start the Discussion by alluding to previous studies in the literature that have shown parental effects on offspring behaviour (second paragraph). However, we now make it clearer throughout that this is not the primary aim of our study. As we now emphasize, our interest is not so much in parental effects on behaviour per se, but in the fitness costs and benefits associated with behavioural traits – and these have seldom been quantified before.

5) A fundamental difference between experiment 1 and experiment 2 seems to be that in the former a set of young (20 or 5) was given to the parents, whereas in the latter the females produced clutch size they wanted. Could this difference affect why female survival was influenced in experiment 1 but not experiment 2?

Yes – and we have now included this possibility in the Discussion.

What drives the difference in the number of offspring produced in experiment 2: different number of eggs produced, or differential provisioning of parental care?

Unfortunately, we did not collect data on clutch size in these experiments, and so we cannot answer this question.

6) The model seems rather divorced from the rest of the paper, and under-emphasized as well. The model should be better integrated with the main flow of the paper.

We have rewritten the Introduction and the Discussion to make it clearer how the model is part of the same broad question that this paper addresses, namely how parental effects influence the costs and benefits that derive from a behavioural trait.

Perhaps an additional figure is needed to illustrate the model schematically.

We have included a new figure as suggested (Figure 4).

*7)*
Figure 5
*does not yet provide a compelling illustration of the sexual conflict result. From this figure it is immediately clear whether the slower rate of increase in biparental parental care with α*_*F*_
*results from the sexual conflict story rather than simply as a result of lower marginal returns on investment given the higher baseline investment level given biparental care.*

We thoroughly agree with this criticism. Consequently we have removed the uniparental care model from the manuscript, and replaced it with a better way to remove conflict. We now model both ‘conflict’ and ‘no conflict’ scenarios assuming biparental care, but in the latter case we specify that mating partners are always genetically identical (*r* = 1). This means that mutants invade in genetically homogeneous groups; these groups however consist of individuals that can differ in phenotype (parental quality). While this option is necessarily biologically somewhat artificial (after all, conflict-free scenarios do not really exist except in truly special cases), our new ‘no conflict’ model does the requested job of entirely removing sexual conflict while retaining all other features of the original model.

*8) The authors need to think carefully about what it means to claim that females exploit high quality males. The authors show in*
Figure 4
*that males suffer from having lower-quality mates. But this is not necessarily exploitation; these lower quality mates could be doing the best they are able and the result would still obtain.*

We now provide a more explicit definition of exploitation in the Introduction, namely that exploitation occurs when the actions of another individual cause fitness loss in a focal individual. By this definition, we have shown exploitation – for the following reasons. The males in this experiment were as alike as they could be, because we used tetrads of brothers across the experimental treatments, and so each had a very similar rearing environment. Therefore any systematic differences in fitness (such as we found) can be attributable to the actions of their partner. Since some males lost more fitness than others, by our definition these individuals are being exploited by their partners.

I would think that at the bare minimum, exploitation would involve females contributing lower levels of parental care when paired with higher quality males. To be clear, I am not asking for additional experiments to test this, just for a reconsideration of what one must observe to demonstrate exploitation.

We emphasise two key points in responding to this point:

A) Formal definitions of exploitation centre on fitness change (see above), not on behavioural change;

B) Measures of behavioural change seldom map perfectly on fitness change (e.g., [43]).

We have avoided these problems by measuring fitness change alone, and avoiding the complication of measuring behavioural traits. Therefore we don’t think there is an alternative interpretation of our data along the lines the referee suggests.

[Editors' note: further revisions were requested prior to acceptance, as described below.]

From Reviewer 1:

The authors have done a good job of differentiating quality from body size. Both effects are included in the analysis and the results are consistent in addition to the size effect. I do think the authors should address the size effect a little more directly. I suggest treating this in the Discussion to make clear to readers that the quality effect is in addition to size effects.

We have added two sentences at the start of the Discussion to acknowledge the effect of size, but we also emphasize that this is not the novel (nor, in our view, the most interesting) finding of our study.

Lifespan as a measure of fitness is OK, but the effect size shown here (i.e., 4-8 days of additional life) seems quite modest.

First, we wonder exactly how modest this contribution is. A male might mate with multiple females repeatedly in 4-8d, thus substantially increasing his reproductive success (Discussion, third paragraph). Second, even small changes in fitness are not trivial on an evolutionary timescale, a case that is made repeatedly in textbooks on Evolution.

I agree that they have demonstrated that potential exploitation of high quality individuals is a risk, but the risk seems low and the cost in terms of lifespan seems low as well compared to the potential benefits of increased brood size and quality.

We do not understand the point the referee is making here: specifically, what do they mean by the potential benefits of increased brood size and quality? These measures do not increase when males are paired with a low quality partner.

In their effort to determine if this sort of pairing (low quality with high quality) happens in nature, they paraphrase a study where carcasses were added to the environment. Half of the carcasses were uncontested in the study, and the authors suggest this provides a conservative estimate. However, by increasing the number of carcasses, the cited study actually causes an effective decrease in density of beetles. Because carcasses are likely less abundant than in this study, most carcasses are likely to be contested under natural conditions. Thus, the authors may be overestimating the likelihood of uncontested carcasses and the coincident possibility of exploitation by low quality individuals.

True – but the local beetle population might have increased too, as more individuals were attracted by the over-abundant carrion. We have added a sentence to make it clearer that the field study is hard to interpret in this context (Discussion, seventh paragraph).

From the Reviewing Editor:

I suggest that you expand the Abstract to better describe the specifics of what you have found in the experiments: (1) receiving high parental care as a juvenile results in individuals having higher success at raising large broods and lower mortality from doing so; (2) high quality males have to compensate for their low-quality partners and thus suffer higher mortality rates than high quality males with high quality partners.

We have added two sentences to the Abstract, modeled on those suggested by the editor.

In the subsection “Outline of model”, please stress that the “sealed bid” in the model is (if I understand correctly) contingent on the type of the partner, not a single bid irrespective of partner type.

This is a good idea. We now write: “This is equivalent to a ‘sealed bid’ model of parental care (22), though in our case the bid is multidimensional (quality-dependent).”

In the Discussion section, the authors state: “Under these conditions, and according to the fitness-based definition of selfish exploitation we are using here (see Introduction), males were exploited by their lower quality partners”. But in the model, I think this happens even when mates are genetically identical. Would you say that they are “exploiting” one another? It is hard to envision why selection would favor true exploitation in that case. Please clarify.

The model predicts that when a low quality individual A partners with a high quality individual B, then A will care less than usual, B will care more than usual, and B will consequently die younger. These qualitative predictions hold both with and without sexual conflict. Importantly, however, when sexual conflict is absent, the evolved reduction in care by A is in the fitness interests of both individuals. This is because B gains indirect fitness benefits from A living longer. Consequently, partner exploitation does not evolve in the ‘no sexual conflict’ model. In the experiments, by contrast, coparents were not close relatives and so there were no indirect fitness benefits when a partner provided less care.

In short: exploitation does not occur in the ‘no conflict’ model because the reduction in direct fitness when a low-quality mate provides reduced care is compensated by indirect fitness benefits due to the mate living longer. This is not possible in the experiments, where the mating pairs are not close relatives.

We have added the following text to the model results to make this clearer:

“However, when coparents are unrelated (as in the experiments and in the model with sexual conflict) a reduction in care by one individual will reduce the fitness of its partner and can consequently be considered exploitative.”

We also now stress how completely conflict really is removed:

“While biologically somewhat artificial, this scenario removes sexual conflict entirely while retaining all other features of the original model (note in particular that the mutants within a group can differ in parental quality despite being genetically identical, as they may have experienced different early life conditions; but differences in behaviour based on phenotype are now more analogous to arms and legs of a single individual human acting differently to aid in the joint task of efficient locomotion, than to a conflict scenario where one part tries to make the other do more work).”